# Chemically Modified Extracellular Vesicles and Applications in Radiolabeling and Drug Delivery

**DOI:** 10.3390/pharmaceutics14030653

**Published:** 2022-03-16

**Authors:** Elisa-Racky N’Diaye, Nicola Salvatore Orefice, Catherine Ghezzi, Ahcène Boumendjel

**Affiliations:** 1Faculty of Pharmacy, Université Grenoble Alpes, 38700 La Tronche, France; elisa-racky.n-diaye@etu-univ-grenoble-alpes.fr; 2Laboratoire Radiopharmaceutiques Biocliniques (LRB), INSERM U1039, Faculté de Médecine La Tronche, Université Grenoble Alpes, 38000 Grenoble, France; nicola.orefice@univ-grenoble-alpes.fr (N.S.O.); catherine.ghezzi@univ-grenoble-alpes.fr (C.G.)

**Keywords:** extracellular vesicles, nanoparticles, drug delivery, radiolabeling

## Abstract

Extracellular vesicles (EVs) have been exploited as bio-inspired drug delivery systems (DDS) in the biomedical field. EVs have more advantages than synthetic nanoparticles: they are naturally equipped to cross extra- and intra-cellular barriers. Furthermore, they can deliver functional biomolecules from one cell to another even far away in the body. These advantages, along with obtained promising in vivo results, clearly evidenced the potential of EVs in drug delivery. Nevertheless, due to the difficulties of finding a chemical approach that is coherent with EVs’ rational clinical therapeutic use, those in the drug delivery community are expecting more from EVs’ use. Therefore, this review gathered knowledge of the current chemical approaches dealing with the conjugation of EVs for drugs and radiotracers.

## 1. Introduction

Over the decades, minimally invasive synthetic drug delivery systems have been engineered to overcome the limitations of free therapeutics and navigate heterogeneous biological barriers across patient populations and diseases, increasingly needing a personalized clinical intervention for therapeutic efficacy. Synthetic drug delivery systems, as nanoparticles, have also been developed to improve the clearance and distribution profile of a therapeutic intervention that is mainly governed by the vehicle’s character rather than by the drug molecule’s physicochemical properties [1]. However, despite the advantages that nanoparticles offer—such as improving stability and solubility of encapsulated cargos, promoting transport across membranes, and prolonging circulation times to increase safety and efficacy for therapeutics delivery [2]—their use is still associated with several drawbacks. Notably, a rapid clearance via the reticuloendothelial system [3], accumulation in the spleen and liver, and acute hypersensitivity reaction represent the target organ dose [4,5]. Amongst the continually expanding area of interest in the field of biological or bioinspired drug delivery systems, that of extracellular vesicles (EVs) has been growing expeditiously [5]. These heterogeneous populations of naturally occurring nano- to micro-sized membrane vesicles, capable of transporting biomolecules from producer to recipient cells [6], have improved our understanding of new forms of cell-cell communication. However, these systems are attractive for turning into commercial products because they have one crucial advantage in common: they come from living cells [7]. Most of the current studies employ a few well-characterized cell lines to produce EVs, including microvesicles (MVs), exosomes, or exosome-like vesicles (ELVs) [8], and notably stem cells that represent a natural choice because they can be cultured long-term and do not produce an immune response. Recently, a study has reported that specific progenitor cell-derived EVs convey biological cargo promoting angiogenesis and tissue repair and modulating immune functions [8,9]. This has drawn particular attention towards applying the EVs for therapeutic delivery to overcome synthetic drug delivery systems-associated issues [10]. Another intriguing aspect of EVs is their intrinsic stability, circulation, and ability to carry and protect a wide array of nucleic acids into recipient cells, avoiding the mononuclear phagocytic system (MPS) by exhibiting surface protein CD47 [11]. It is also significant to highlight that EVs can comprise proteins that bind to and sort their RNA [12]. Therefore, EVs represent a promising source for engineering systems to deliver therapeutics under different clinical conditions such as cancer medicine, immunotherapy, and in vivo gene editing. In this regard, genetic modifications of EV-secreting cells have been applied to target restricted cellular receptors. However, this genetic engineering of EV-donor cells appears to be cumbersome and time-consuming. Unfortunately, engineered EV-donor cells are not adequately exposed or adequately stable to act as an efficient drug delivery mechanism that includes multiple steps of cloning, transfection, viral transduction, selection, and large-scale cell culture and EVs purification.

Furthermore, although it can enable stable conjugation of EVs with targeting moieties, genetic manipulation poses a high risk of horizontal gene transfer because the process may incorporate high-copy plasmids or transgenes that are eventually transferred to target cells. Even though different approaches requiring genetic modification of EV-secreting cells have been applied to overcome these complications, they still suffer from the same limitations described above for EV-donor cells. Therefore, a series of alternative methods addressing the modification of EVs after secretion without manipulating the EV-producing cells are still needed to avoid genetic manipulation. In this regard, recent studies have provided evidence that click chemistry can be efficiently used to modify EV-producing cells [13] or purified EVs [14] to generate “tailored” vesicles.

This review focuses on improving radio imaging and understanding EVs’ physiological/pathophysiological behavior before using them as drug carriers. We also report how chemically modified EVs can enhance drug delivery, facilitate clinical translation of precision medicines, improve EVs-based precision therapies, and overcome biological barriers and patient heterogeneity.

## 2. Chemical Modification for In Vivo Tracking Extracellular Vesicles

Several limitations for in vivo tracking of EVs, such as poor penetration depth and spatial resolution, make it unsuitable for their complete clinical translation. Nuclear medicine imaging could be a good option for tracking EVs and evaluating their biodistribution. This method provides three-dimensional images using single-photon emission computed tomography (SPECT) or positron emission tomography (PET). Furthermore, nuclear imaging combined with anatomical imaging, such as computed tomography (CT) or MRI, represents a good option for providing better tracking of the localization of the EVs. This approach provides excellent sensitivity and more straightforward quantification, making their clinical applications feasible. However, it is equally important to stress that one significant restriction of this nuclear imaging technology is the possibility of altering EVs proprieties by the transduction procedure. EV subpopulations, such as exosomes and microvesicles, have physiological properties suitable for radio imaging. Indeed, the primordial role of EVs is in long-distance cell–cell communication because the secreted EVs can enter circulation and pass through additional biological barriers, making them suitable for real-time monitoring in their native environments [9,10]. An efficient strategy for EVs conjugation with a radionuclide could provide an enhanced understanding of EVs’ functions in the physiology and pathophysiology of many diseases. Likewise, the characterization of their pharmacokinetics and biological behavior could be constructive for fostering improved diagnoses and treatment of many pathologies. Because nuclear imaging modalities can also provide information about the therapeutic dose of EVs and their potential side effects [15], definite chemical modifications are needed within the EVs research field to overcome the radiolabeling-associated drawbacks and enhance their use for in vivo tracking.

### 2.1. Covalent-Binding Method

Click chemistry is a novel approach for conjugating peptides, antibodies (Abs), or even fluorescent and radioactive agents on the surface of an EV. This conjugation of extracellular moieties could enable a specific interaction with a target cell. The click chemistry reaction corresponds to a copper-catalyzed azide–alkyne cycloaddition between an alkyne and an azide, providing a triazole linkage (Figure 1). Since click reaction is known to be potentially cytotoxic because of its copper-catalyzed azide–alkyne cycloaddition (CuAAC) [16], significant effort has been dedicated to developing a catalyst-free variant of azide–alkyne click chemistry. 

The necessity to conjugate peptides or radioactive agents on the surface of EVs while avoiding cell toxicity has led to a novel method based on a copper-free biorthogonal cycloaddition [17]. The strain-promoted azide–alkyne click (SPAAC) chemistry enables a triazole linkage after the reaction between an azide and a strained alkyne (cyclooctyne) (Figure 1) [18,19]. 

In the field of EVs, the bioorthogonal SPAAC method was based on azide-containing sugars incorporated into the glycoproteins at the surface of exosomes. Hence, tetra-acetylated *N*-azidoacetyl-D-mannosamine (Ac4ManNAz) was incorporated into glycans inside the cells. This Ac4ManNAz is later redistributed into their exosomes, making them azido-containing exosomes. Then, a biorthogonal click reaction with aza-dibenzyl-cyclooctyne (ADIBO)-fluorescent dyes was employed to label those exosomes (Figure 2). The in vivo biodistribution of the azido-containing exosomes, explored in Cy.5.5 exosomes, originated from MCF7 and MDA-MB-231 cells and administered to tumor-bearing mice, detected labeled exosomes distribution more in tumors than in blood and muscles and accumulated in the liver and intestines. 

One of the alternatives to the click chemistry approach is the incorporation of radioactive technetium (^99m^Tc) on the EVs membrane surface. Hwang et al. radiolabeled macrophage-derived exosome-mimetic nanovesicles (ENVs) with ^99m^Tc-HMPAO to observe the biodistribution of EVs in vivo [20] (Figure 3).

^99m^Tc-HMPAO is often used in the clinic for in vivo imaging. The ^99m^Tc-HMPAO reacts with the sulfhydryl groups of the glutathione naturally present in cells and becomes a hydrophobic compound. After being trapped in the cell, ^99m^Tc-HMPAO migrates on the EVs lipid bilayer [21]. To obtain ^99m^Tc-HMPAO, hexamethyl-propylene-amine-oxime (HMPAO) reacted with SnCl_2_, ^99m^TcO_4_, and NaCl under appropriate conditions. As a result, the ^99m^Tc-HMPAO-ENVs became smaller-sized EVs for a small fraction, suggesting that the radiolabeling could disaggregate EVs (Figure 4). However, the size of most ^99m^Tc-HMPAO-ENVs after the radiolabeling remained unchanged. Furthermore, a Western blot analysis showed that the concentration of exosome-specific proteins, such as CD63, did not change despite the radiolabeling.

As far as biodistribution is concerned, ^99m^Tc-HMPAO-ENVs accumulates in the liver and the spleen for 30 min after injection. After three h, a little radioactive signal was detected in the salivary glands. Interestingly, no biodistribution was seen in the brain, while a high signal was seen for ^99m^Tc-HMPAO-only particles [21]. In this context, Varga et al. evaluated erythrocyte-derived EVs’ biodistribution under SPECT/CT [22]. The EVs, radiolabeled by ^99m^Tc-tricarbonyl complexes, were injected and underwent SPECT imaging, which showed an accumulation of the ^99m^Tc-Exos, mainly in the liver and spleen. It was reported that only a minor portion of ^99m^Tc detached from the exosomes and could be seen in the bladder [22]. However, the methods employed [20,22] required commercial kits with more expensive and complex radioactive precursors. Moreover, they imply longer reaction times and more complex chemistry to incorporate the radionuclide [23]. Therefore, to avoid taking too much capital on the method and to emphasize better results, they set up a radiochemical radiolabeling of milk-derived exosomes (MDE) with reduced ^99m^Tc (IV). The reduction reaction of ^99m^Tc occurred under an N_2_ atmosphere for 5 min at 37 °C and was neutralized with NaOH, and the final ^99m^Tc(IV) was employed in radiolabeling of MDE. To check whether a difference in the radio efficiency could exist between reduced ^99m^Tc-MDE and commercial ^99m^Tc-MDE, they prepared sodium pertechnetate (VII) with acetic acid under the same conditions. This surface labeling provoked no modification in the EVs’ biological and physiological properties with a radio efficiency of 99.5%.

The different types of radiolabeled MDEs were administered intravenously (IV), intraperitoneally, and intranasally in mice to see whether the route of administration could impact the biodistribution. Following SPECT/CT imaging, it was found that after the IV administration, reduced ^99m^Tc-MDE accumulated rapidly in the liver and the urinary bladder (fast urinary excretion) and was distributed in the aorta and lungs (quick blood clearance). After the intraperitoneal administration, reduced ^99m^Tc-MDE was distributed mainly in the abdominal cavity, spleen, and thyroid. The intranasal route of administration provoked a biodistribution in the nasal cavity, trachea, and lungs, whereas SPECT/CT could not detect any signal in the brain. The autoradiography showed that reduced ^99m^Tc-MDEs were 10-fold more present in the liver than in the brain after the IV administration. After the intraperitoneal administration, the autoradiography showed two times more reduced ^99m^Tc-MDEs in the liver than in the brain [23]. These two studies showed that the chemical radiolabeling of EVs enables the characterization of their pharmacokinetics.

Because SPECT has many advantages, such as high sensitivity, good spatial resolution, and limitless penetration depth, Royo et al. used PET/CT imaging to determine whether the glycosylation modification could impact the biodistribution of mouse liver proliferative cell-derived EVs in mice [24]. This hypothesis was emphasized based on studies reporting that glycosylation of EVs surface can control their interaction in cell-to-cell communication [25]. EVs were treated with neuraminidase, an enzyme that digests terminal sialic acid residues from glycoproteins to test this hypothesis. To check the difference in the biodistribution due to the glycosylation, neuraminidase-treated (Neu-EVs) and untreated EVs (EVs-only) were labeled with ^124^I. Hence, the EVs were administered via two different routes of administration (IV and injection in the hock of the mice) to assess the variation in the biodistribution. It appears that both Neu-EVs and EVs were distributed more in the liver and lung within the 15 min after their administration and sensibly less in the thyroid gland. However, 15 min after their administration, the biodistribution in all organs decreased except in the thyroid and the bladder, where it increased. After a *t*-test, a statistical difference was shown between the distribution of the Neu-EVs and the EVs just at the last time point. It was concluded that the biodistribution of EVs could be modified because of the glycosylation reaction. This feature could be beneficial in targeted drug delivery systems, where a chemical modification could influence the distribution of the drug and its reach to targeted sites (Table 1).

### 2.2. Bifunctional Chelators for Membrane Radiolabeling

An alternative method for improving our knowledge of EVs’ in vivo behavior is conjugating bifunctional chelators (BFC) at their surface (Figure 5). A bifunctional chelator is a two-parted molecule with one functional group on the one hand and a metal-binding moiety. The active group allows a covalent attachment with amines, thiols, or carboxylic groups at the surface of EVs, whereas the metal moiety offers radionuclide sequestration. The radionuclide used in this method is ^64^Cu, ^68^Ga, or ^11^N [26]. 

The BFC is attached to EVs via click chemistry. In this context, Faruqu proposed a novel, reliable, and universal method for the radiolabeling of exosomes [27]. Melanoma (B16F10)-derived exosomes (EXO_B16_) were labeled in two ways: an intraluminal labeling and ^111^Indium-chelated labeling, aiming to identify which EVs labeling was the most efficient. The chelated EXO_B16_ was obtained after reacting with diethylenetriaminepentaacetic dianhydride (DTPA-anhydride) and dry-chloroform added to the EXO_B16_. Finally, ^111^InCl_3_ in an ammonium buffer was added. Both labeled exosomes were injected intravenously in melanoma-bearing competent and immunodeficient mice. Whole-body SPECT/CT was collected, and mice were sacrificed for ex vivo gamma counting. It appears that ^111^In-DTPA-EXO_B16_ had a better radiolabeling efficiency and radiochemical stability than intraluminal labeled exosomes. SPECT/CT showed the presence of ^111^In-DTPA-EXO_B16_ first in the liver and spleen for >24 h, and later in the bladder, but no signals were detected in tumors. The gamma-counter showed a rapid blood clearance of ^111^In-DTPA-EXO_B16_, an increasing presence in the tumor, and a low urine excretion compared with free-^111^In urine excretion, whereas feces excretion remained similar. Unfortunately, the signal ex vivo remained too low to conclude that ^111^In-DTPA-EXO_B16_ could properly reach tumors. This novel method is correctly adapted for live imaging and quantitative biodistribution for all types of exosomes but is still inefficient for drug delivery since exosomes do not reach tumors [27]. In this research, it is essential to note that gamma-counting could detect signals ex vivo, which SPECT could not. The less sensitivity and noise in SPECT make researchers displeased, encouraging them to work with BFC but PET instead. Banerjee and colleagues reported a two-step surface modification method of small EVs (SEVs) with ^64^Cu^2+^ for PET/MRI imaging [28]. In this study, human umbilical cord blood mononuclear cell-derived SEVs (hUCB-MNC SEVs) free thiol group bound with dodecane tetraacetic acid (DOTA) maleimide group. The DOTA-SEV was complexed with ^64^CuCl_2_ to obtain the bifunctional chelator as a radiological marker. An ex vivo radioactivity quantification was made after mice sacrifice to show the biodistribution of the new hUCB-MNC SEVs. As a result, signals were detected in the following order (from the highest signal to lowest): liver > lungs > kidney > stomach > brain. These results were examined by PET/MRI and indicated a soft but homogenous presence of hUCB-MNC SEVs in the brain, in the striatum, prefrontal cortex, and the cerebellum [28]. This remarkable capacity to pass the blood–brain barrier (BBB) allows supposing that this method is appropriate for the in vivo tracking of EVs in low accumulation organs such as the brain. Based on the BFC strategy, Jung et al. searched for an acceptable imaging method for EVs’ in vivo tracking [29]. Herewith, PET, optical imaging, and ex vivo radioactivity quantification were used to see which one was more performant for tracking 4T1-exosomes (EXO) coupled to copper or gallium-BFC. The exosomes were either radiolabeled with bifunctional chelation between ^64^Cu or ^68^Ga NOTA in the presence of sodium carbonate buffer (pH 9.5), and the EVs surface proteins’ amino group was fluorescently labeled. The exosomes were injected via the lymphatic or hematogenous route to study pharmacokinetics. It was found that each labeled exosome remained stable and conserved its final size. Thus, PET imaging could detect the biodistribution of the BFC-4T1-EXOs with more detail than optical imaging. After the lymphatic administration, BFC-4T1-EXOs had a greater uptake in the lymph nodes than the BFC-only or free-Cu, more specifically in the brachial or axillary lymph nodes. No radioactive signal was detected in inguinal lymph nodes or any other organs after the lymphatic route of administration. In contrast, greater uptake in the lungs, liver, and spleen was identified after the hematogenous course. The same biodistribution results were found after ex vivo radioactivity quantification or with ^68^Ga-NOTA-4T1 EXOs. This study concluded that PET is an acceptable imaging method for the in vivo tracking of EVs and that ^64^Cu and ^68^Ga are excellent candidates for clinical application [29]. The next following method is based on the radiolabeling of PEG-conjugated exosomes. PEGylation of EVs is known to improve their pharmacokinetics, allow a more significant accumulation in tumors, and decrease their premature hepatic sequestration and clearance [30]. Shi et al. studied in vivo with PET imaging some ^64^Cu-PEG-modified exosomes after stable chelation [31]. 4T1 breast cancer-derived exosomes were conjugated to amine-reactive NOTA-^64^Cu bifunctional chelator and PEG. The addition of PEG neutralized the surface charge of exosomes and slightly increased their size, but no additional changes were found. The radiolabeling method showed high stability and took a brief time (1 min). Next, ^64^Cu-NOTA-Exos and ^64^Cu-NOTA-PEG-Exos were injected in 4T1 tumor-bearing mice. ^64^Cu-NOTA-Exos had a very short blood circulation and a full hepatic clearance. High signals that were detected in the liver decreased over time, and low tumor uptake and poor tumor contrast were identified. Surprisingly, ^64^Cu-NOTA-PEG-Exos had a prolonged blood circulation, a reduced hepatic clearance, and a liver uptake lower than ^64^Cu-NOTA-Exos. It should be highlighted that ^64^Cu-NOTA-PEG-Exos accumulated three times more in tumor than ^64^Cu-NOTA-Exos with an enhanced tumor contrast [31]. This efficient PEGylation method provides an exciting improvement in the pharmacokinetics of EVs, which represent an advancement in tumor-targeted drug delivery systems (Table 2).

## 3. Chemical Modifications on Extracellular Vesicle-Mediated Delivery Cargo

EVs have emerged as a powerful tool for drug delivery, including their intrinsic homing ability, biocompatibility, cell-specific targeting, non-immunogenicity, broad distribution in biological fluids, and easy penetration across physiological barriers [6,7]. However, one of the significant limitations of EV-based drug delivery has been the lack of efficient isolation methods. In particular, conventional EV isolation techniques have limited yields, low purity, and inadequate batch-to-batch consistency. Therefore, chemical modifications have been developed to exploit EV drug delivery potential, introduce and stabilize the cargo of exogenous origin into EVs, and maximize their efficacy of targeting and delivery. Here, we discuss the chemical strategies employed for the EV cargo loading, targeting, and unloading.

### 3.1. Covalent Binding Approach

Click chemistry can also improve the intracellular delivery of therapeutic EV cargo. Click chemistry is used to alter the character of EVs surface, as described in Section 2.1. Smyth et al. tested whether the linkage of azide-fluor 545 on the surface of an EV would change its function [14]. For such purpose, first, exosomes derived from 4T1 breast cancer cells were functionalized with a terminal alkyl group. Then, the amine group present on the 4T1 derived EVs’ surface was cross-linked with the carboxyl group of a 4-pentynoic acid using carbodiimide activation [14]. This enabled the conjugation of the 4T1-derived EVs with azide-fluor 545 thanks to click chemistry. It was reported that the chemical modification impaired no modification of the natural functions of the EV, and as expected, the copper catalyst was potentially cytotoxic [10]. 

Nonetheless, click chemistry tends to be time-consuming and requires reaction conditions that are difficult to master [32,33,34]. Those limitations motivated investigators to conjugate an aptamer on the surface of EVs using covalent binding with another method than click chemistry. This method was applied in improving the delivery of the anticancer drug paclitaxel to target cancer cells [33]. This innovative method covalently modifies the surface of a dendritic cell-derived EVs loading paclitaxel (PTX). The surface modification showed a 6-fold and 3-fold treatment efficacy in vitro and in vivo, respectively, compared with unmodified PTX-loaded EVs. Moreover, the added cholesterol could also confer the EVs better rigidity and stability by enhancing the hydrophobic–hydrophobic interactions in lipid bilayers [35]. Finally, it was claimed that a significant amount of EVs could be prepared in approximately one hour. All these advantages favor the clinical translation of this method in the future. 

Similarly, it was suggested to study a permanent covalent bond between peptides or specific nanobodies and EVs’ surfaces. Thanks to a simple enzymatic method on EVs targeting several cancer cells, this bond was possible. EVs with either an epidermal growth factor receptor (EGFR)-targeting peptide or anti-EGFR nanobody improved their accumulation in EGFR+ cancer cells. This occurs in vitro as well as in vivo [36]. Interestingly, this enzymatic method using protein ligases is also efficient on EVs with peptides and nanobodies targeting other receptors. This method is not specific to a defined type of receptor. It could be worthwhile to know whether this method could apply with a more prominent protein of interest. Moreover, the modified EVs could also efficiently deliver paclitaxel or RNA to cancer cells (Table 3).

Hence, it can be concluded that the modeling of covalent bonds on EVs could allow their use not only in preclinical stages but potentially in humans.

### 3.2. Non-Covalent Binding

Two main non-covalent binding methods were reported. The first one consisted of the bond between EVs and surface peptides thanks to electrostatic interaction. An alternative approach was proposed to influence the delivery of exosomes with magnetic strength. Hence, Nakase and Futaki explored a simple technique for enhancing exosomes’ cellular uptake and cytosolic release [37]. They combined a pH-sensitive fusogenic GALA peptide with a commercially available cationic lipid: lipofectamine (LTX). The electrostatic interaction occurred between the positively charged LTX and the negatively charged surface membrane of a CD63-green fluorescent protein (GFP)-tagged exosome. To study the cellular uptake, HeLa and CHO-K1 cells were treated with the GFP-GALA-Exos with and without the addition of lipofectamine. It was found that 4% of LTX increased the cellular uptake of GFP-GALA-Exos 15-fold by HeLa cells and 175-fold by CHO-K1 cells. Unfortunately, a higher LTX concentration could induce cytotoxicity.

Thus, an alternative was studied to check whether a lower dose of added LTX could always increase the cellular uptake, and indeed it did. The addition of 2.0% LTX increased the cellular uptake by six-fold. This encouraging result led to encapsulating dextran in GFP-GALA-LTX-Exos, and the result was an increase in cellular absorption and drug release [37]. Tamura and colleagues worked on EVs whose surface was modified with cationized pullulan (commonly named pull+) [38]. Pullulan can target hepatocyte asialoglycoprotein (ASGPR) receptors [39], and this property will help exosomes conjugated with cationized pullulan reach injured liver sites. The +pull-Exos were easily internalized in HepG2 cells, reflecting an excellent cellular uptake. After systemic administration in mice with concanavalin A-induced liver injuries, +pull-Exos were distributed readily in the liver. 

The necrotic areas were at their lowest in these same regions, which shows an enhanced anti-inflammatory effect of +pull-Exos [38]—adding cationic agents positively impacted drug delivery. It would still be necessary to monitor the concentration of these cationic agents to avoid and prevent cytotoxicity.

Concerning the magnetic method, Maguire et al. managed a study on using streptavidin-conjugated magnetic beads to influence the targeting of a new kind of microvesicles [40]. During the production of adeno-associated viruses (AAV), it seems that they are naturally associated with nearby exosomes and form so-called vexosomes [40,41]. These new nanoscale vehicles are less immunogenetic and more biocompatible than normal AAVs. Those mRNAs containing vexosomes were bound to magnetic beads to see whether they could react to the attraction of a magnetic field. Hence, small magnets adhered to one region of the numerous well plates that were used. The strategy was applied to biotin acceptor peptide transmembrane domain (BAP-TM) receptors to be incorporated by the vexosomes to allow their specific cell targeting and eventual binding to biotinylated ligands via a streptavidin bridge. This streptavidin bridge then reacted with the streptavidin-conjugated magnetic beads. It was found that after activation of the magnetic field, two times more vexosomes joined the magnetic region, suggesting a more specific targeting by the streptavidin-conjugated magnetic exosome when biotinylated ligand was expressed on the microvesicles surface [40]. With the encouraging in vitro results, this promising method needs to be confirmed in vivo. In this context, Qi et al. carried out an in vivo study of blood-derived exosomes endowed with magnetic properties as a new targeted drug delivery system in cancer therapy [42]. Hence, they developed a dual-functional reticulocyte-derived exosome-based superparamagnetic nanoparticle cluster (SMCNC-Exo) through transferrin conjugated SMCNCs bound to the transferrin of reticulocyte-derived exosomes. The SMCNC-Exo was loaded with doxorubicin via hydrophobic effects. This drug-loaded SMCNC-Exo (D-SMCNC-Exo) was described as biocompatible for drug delivery.

Regarding in vitro drug release, at pH 7.4, approximatively 80% of doxorubicin was released after 8 h. Concerning the in vivo biodistribution in hepatoma 22 subcutaneous cancer-bearing mice, after applying a magnetic field (MF), D-SMCNC-Exos were 1.7-fold more at the cancer site than without the MF. Doxorubicin-SMCNC-Exo succeeded in slightly inhibiting the growth factor without the help of a magnetic field. Still, the entire suppression of the tumor growth factor was possible only under MF [42]. This study demonstrates the very promising progress in EV-inspired drug delivery and its application in cancer therapy (Table 4).

### 3.3. Hydrophobic Insertion

EVs’ last chemical modification method concerns the insertion of hydrophobic molecules on the membrane of exosomes or exosome-like vehicles. As the membrane of EVs is made of a phospholipid bilayer, it is possible to modulate this property to improve the use of EVs as drug delivery vehicles. In this context, Kim et al. developed an in vivo study of the engineering of macrophage (stemming from the primary bone marrow)-derived exosomes for targeted paclitaxel delivery to pulmonary metastases [43]. Paclitaxel-loaded macrophage-derived exosomes with incorporated aminoethylanisamide-PEG (AA-PEG) could bind specifically to the sigma receptors overexpressed in lung cancer cells; aminoethylanisamide is a ligand of sigma receptors. It seems that after the injection in mice with pulmonary metastases, AA-PEG-PTX-Exos showed greater antineoplastic efficacy than Taxol or PTX-Exos. Furthermore, the modulated exosome provides the eradication of pulmonary metastasis because of the high inhibition of tumor growth of AA-PEG-PTX-Exos. This innovative method based on aminoethylanisamide-PEG hydrophobic insertion improved the loading capacity of paclitaxel and its accumulation in cancer cells upon systemic administration, and it is a more excellent therapeutic outcome [43].

The second study concerns the hydrophobic insertion of cholesterol to improve exosome-based cancer therapy’s therapeutic effects [44]. Their key feature of the method relies on an RNA aptamer–protein interaction after the loading of anticancer molecules by a reversible light-inducible protein-protein interaction and the remodeling of the exosome’s producer cells (Table 5). Thus, AS1411 aptamer modified the surface of exosomes because of its hydrophobic membrane, which can interact with cholesterol. This hydrophobic insertion induced a good internalization of the exosomes in K562 leukemia cells. In addition, the AS1411-Exos contained microRNA-21 sponges, which are inhibitors of miR21 in K562 cells, contributing to cancer initiation, progression, and metastasis [44,45]. In the latter study’s frame, the successful delivery of AS1411-miRNA21-Exos was translated by significant inductions of cellular apoptosis [44].

## 4. Discussion

In pharmaceutical science, it is challenging to find the best way to deliver a drug correctly to the organism. There is a need to reach several features that include improving the pharmacological activities, enhancing solubility in aqueous media, enhancing low bioavailability, and increasing the specificity for a target [46]. These features become more challenging when complex organisms such as the brain or the retina are targeted [3]. Hence, extensive research has been carried out for many decades to engineer innovative drug delivery methods. Even if nanoparticles, pro-drug approaches, or viral vectors as therapeutic vehicles showed great promise [7,46], their toxicity, bioavailability, and target delivery put a brake on their applicability [6]. In this context, EVs have emerged as a promising alternative regarding drug delivery. Click chemistry has rapidly emerged as a popular and dominant method for modifying extracellular vesicles chemically. It is a quick and efficient method for improving drug delivery and molecular imaging. Despite its advantages, the major drawback of click chemistry is the toxicity of the copper necessary for use as a catalyst for the copper-catalyzed azide–alkyne cycloaddition reaction (CuAAC) [16]. Therefore, significant effort has been dedicated to developing a catalyst-free variant of azide-alkyne click chemistry. Indeed, copper complexes can be (at a specific dosage) toxic and can negatively affect cellular metabolism and uptake. All of this has consequences on their functionality as catalysts for copper-catalyzed cycloaddition. Copper toxicity is attributed to the oxidative damage caused by reactive oxygen species (primarily hydroxyl radicals or alkoxyl radicals). The toxicity of CuAAc depends on the copper complexes employed and the ligand environment. However, for almost all types of copper complexes, it was shown that the toxicity of CuAAC alters more hepatic cells than other cell lines [47,48,49].

However, these unfortunate events did not stop click chemistry thanks to many copper-free methods [22,40]. The possible fusion of AAV with exosomes could further stimulate the use of AAV as drug carriers in medicine. Indeed, it would mean that exosomes could indirectly improve the use of viral vectors as DDS because of the decreasing immunogenicity and the increase in the biocompatibility they provide. In the imaging area, it is commonly known that SPECT gives less practical information than PET. The remaining question is whether we can use the technique developed under SPECT and PET. Would a simple change in the radionuclide make it possible? Finally, EVs in molecular imaging and DDS could start something big in precision medicine. Therefore, additional investigations are necessary to further our understanding of EVs.

## 5. Conclusions

Nowadays, the study of EVs has become popular in molecular technology research. The precise roles of EVs produced by various cells are still unclear and need further investigation. However, it is essential to know the pharmacokinetics and biodistribution of EVs before their application as DDS. This knowledge can be provided by the in vivo tracking of those vehicles to estimate their behavior in the organism and find ways to enhance their delivery and target capacities. Furthermore, the direct chemical labeling of EVs can positively improve their use in molecular imaging and as drug carriers. 

This review has the merit of being the first collection of specific chemically modified studies of EVs, seldom encountered in either multiple original articles or reviews. Despite a strong focus on elucidating distinct aspects of these vesicles—such as biogenesis pathway and loading nucleic acid drug [50,51], interactions with cells [52], and different modification approach addressed for the design of personalized EVs as DDS [53,54]—there is still much work to be accomplished around integrating the multifaceted capabilities of EVs into DDS. Undoubtedly, the results of the studies gathered in this review demonstrate that manufacturing and administering EVs is feasible and safe, suggesting that the translation of EVs into a therapeutic platform may already be just beyond the horizon. 

In conclusion, it is time to translate the lessons from these preclinical imaging initiatives to the non-human primates (NHPs) imaging community. This will boost the growing EVs’ imaging field by sharing standardized data across multiple sites and studies and allowing meta-analyzes on data that cannot be acquired in the preclinical individual animal laboratory.

## Figures and Tables

**Figure 1 pharmaceutics-14-00653-f001:**
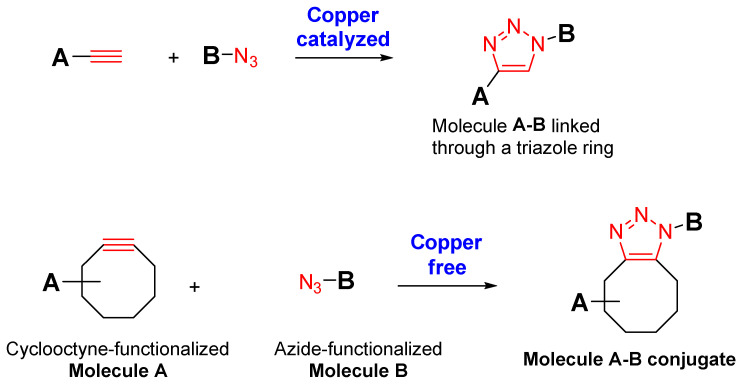
Overview of click chemistry reaction and SPAAC.

**Figure 2 pharmaceutics-14-00653-f002:**
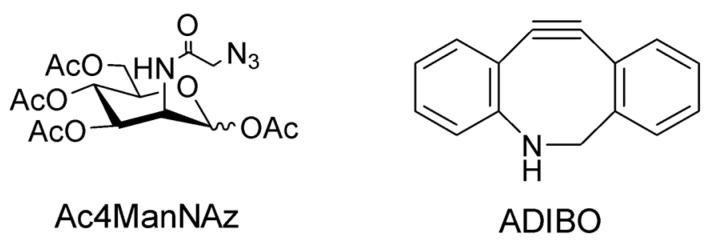
Structures of peracetylated *N*-azidoacetyl-D-mannosamine and aza-dibenzyl-cyclooctyne used in the bioorthogonal SPAAC method.

**Figure 3 pharmaceutics-14-00653-f003:**
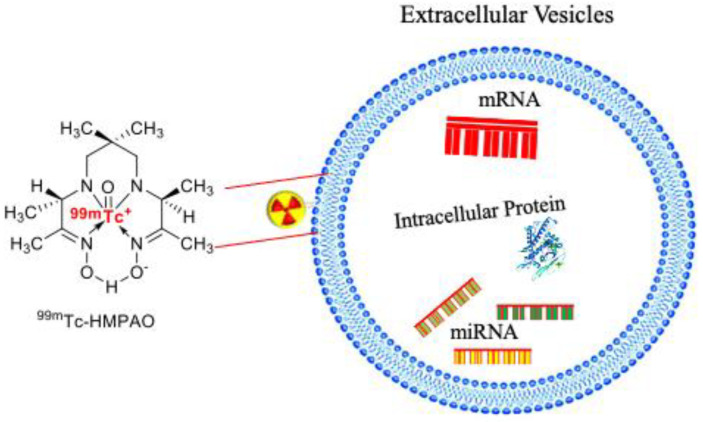
A schematic overview of radioactive technetium (^99m^Tc) on the EVs membrane surface.

**Figure 4 pharmaceutics-14-00653-f004:**
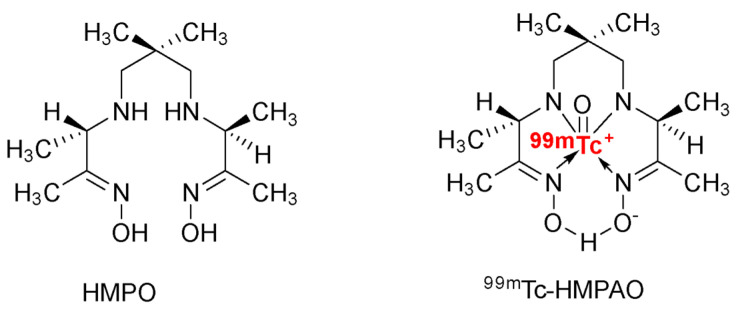
Hexamethylpropylene amine oxime (HMPAO) and its technetium radiolabeled derivative ^99m^Tc-HMPAO.

**Figure 5 pharmaceutics-14-00653-f005:**
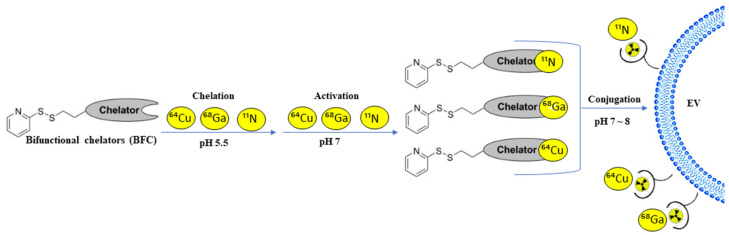
A schematic overview of conjugating bifunctional chelators (BFC) on the EVs membrane surface.

**Table 1 pharmaceutics-14-00653-t001:** Summary of the studies dealing with a chemical modification for in vivo tracking of extracellular vesicles using the covalent-binding method.

Source of Exosomes	Purpose	Method	Results	References
Cy.5.5 exosomes derived from MCF7 and MDA-MB-231 cells	In vivo biodistribution of the azido-containing exosomes	Cooper-free click chemistry with a strain-promoted azide–alkyne click (SPAAC)	Better distribution of the labeled exosomes in tumor than blood and muscles; accumulation in the liver and intestines	[17]
Macrophage-derived exosome-mimetic nanovesicles (ENVs)	Biodistribution of ENVs in vivo	Incorporating ^99m^Tc on the EVs membrane surface with click chemistry	^99m^Tc-HMPAO-ENVs accumulates in the liver, spleen, salivary gland	[20]
Erythrocyte-derived EVs	Erythrocyte-derived EVs’ biodistribution under SPECT/CT	Radiolabeling by ^99m^Tc-tricarbonyl complexes with click chemistry	Accumulation of the ^99m^Tc-Exos in the liver and spleen	[22]
Milk-derived exosomes (MDE)	A cheaper method with higher efficiency to study EVs biodistribution	Radiochemical labeling of MDE with reduced ^99m^Tc (IV) injected intravenously, intraperitoneally, and intranasally	IV: reduced ^99m^Tc-MDE accumulated in the liver and urinary bladder and distributed in aorta and lungs IP: reduced ^99m^Tc-MDE distributed in the abdominal cavity, spleen, and thyroid IN: Biodistribution in the nasal cavity, trachea, and lung	[23]
Mouse liver proliferative cell-derived EVs	Impact of glycosylation modification on the biodistribution of EVs in mice	EVs were treated with neuraminidase and labeled with ¹²⁴I	Distribution primarily in liver and lung and slightly in the thyroid gland	[24]

**Table 2 pharmaceutics-14-00653-t002:** Summary of the studies dealing with chemical modification for the in vivo tracking of extracellular vesicles using bifunctional chelators for membrane radiolabeling.

Source of Exosomes	Purpose	Method	Results	References
Melanoma (B16F10)-derived exosomes (EXOB16)	A novel, reliable, and universal method for the radiolabeling of exosomes	^111^Indium-chelated labeling of EV	Better radiolabeling efficiency and radiochemical stability Distribution in liver, spleen, and bladder	[27]
Human umbilical cord blood mononuclear cell-derived small EVs (hUCB-MNC SEVs)	Biodistribution of the new hUCB-MNC SEVs showed by PET/MRI	2-step surface modification method of small EVs with ^64^Cu^2+^	Biodistribution in liver > lungs > kidney > stomach > brain (striatum, prefrontal cortex, and the cerebellum)	[28]
4T1 breast cancer-derived exosomes	Adequate imaging method for the in vivo tracking of EVs between PET, optical imaging, ex vivo radioactivity quantification	Exosomes were either radiolabeled with a BFC-^64^Cu or -^68^ Ga or fluorescently labeled	PET imaging and ex vivo radioactivity quantification could see the biodistribution of the BFC-4T1-EXOs with more detail than optical imaging	[29]
4T1 breast cancer-derived exosomes	Impact of PEGylation of EVs on their pharmacokinetics	Radiolabeling of PEG conjugated Exosomes	The efficient PEGylation method provides an exciting improvement in the pharmacokinetics of EVs, even in the tumor	[30]

**Table 3 pharmaceutics-14-00653-t003:** Summary of the studies dealing with chemical modifications of extracellular vesicle-mediated delivery cargo using covalent binding approaches.

Source of Exosomes	Purpose	Method	Results	References
4T1 breast cancer-derived exosomes	See whether the linkage of azide-fluor 545 on the surface of an EV would change its function	4T1 EXOs were functionalized with a terminal alkyl group after click chemistry	No modification of the natural functions of the EV was impaired by being chemically modified	[14]
Dendritic cell-derived EVs	Improving the delivery of paclitaxel to target cancer cells	Conjugation of an aptamer on the surface of EVs using covalent binding	The surface modification showed a 6-fold and 3-fold treatment efficacy in vitro and in vivo	[33]
Human red blood cells (RBCs) as a source of EVs	Study of a permanent covalent bond between peptides or specific nanobodies and EVs’ surfaces	Simple enzymatic method on EVs targeting several cancer cells	Epidermal growth factor receptor (EGFR)-targeting peptide or anti-EGFR nanobody improved their accumulation in EGFR+ cancer cells	[36]

**Table 4 pharmaceutics-14-00653-t004:** Summary of the studies dealing with chemical modifications on extracellular vesicle-mediated delivery cargo using non-covalent binding approaches.

Source of Exosomes	Purpose	Method	Results	References
CD63-GFP-containing exosomes derived from HeLa cells and Chines Hamster Ovary (CHO)-K1 cells	A simple technique for enhancing exosomes cellular uptake and cytosolic release	Electrostatic interaction between a positively charged lipofectamine and the negatively charged surface membrane of an EV	LTX increased the cellular uptake of GFP-GALA-Exos 15-fold by HeLa cells and 175-fold by CHO-K1 cells	[37]
Mesenchymal stem cells (MSC)-derived exosomes	Reach injured liver sites	EVs surface modified with cationized pullulan	Excellent cellular uptake in HepG2 cells and good distribution in the liver Enhanced anti-inflammatory effect of +pull-MSC Exos	[38]
Vexosomes are formed by the natural association between adeno-associated viruses and exosomes	Influence of magnetic beads on the targeting of vexosomes	Vexosomes were bound to streptavidin-conjugated magnetic beads	After activation of the magnetic field, two times more vexosomes joined the magnetic region	[40]
Reticulocyte-derived exosomes (REXOs)	Study of a new targeted drug delivery system	Transferrin conjugated superparamagnetic nanoparticle cluster bound to the transferrin of REXOs loaded with doxorubicin via hydrophobic effects	The entire suppression of the tumor growth factor was possible only under MF	[42]

**Table 5 pharmaceutics-14-00653-t005:** Description of the studies dealing with chemical modifications on extracellular vesicle-mediated delivery cargo using hydrophobic insertion approaches.

Source of Exosomes	Purpose	Method	Results	References
Primary bone marrow stemmed macrophage-derived exosomes	Targeting of paclitaxel delivery to pulmonary metastases for systemic administration	Incorporation of amino-ethylanisamide-PEG on the surface of EXOs allows the bond of the sigma receptors to lung cancer cells	Greater antineoplastic efficacy, high inhibition of tumor growth, and better survival time after systemic administration	[43]
Plasma-derived exosomes containing miRNA21	Hydrophobic insertion of cholesterol to improve the therapeutic effects of exosome-based cancer therapy	Modification of loaded exosomes with the hydrophobic insertion of AS1411 aptamer interacting with proteins after a reversible light-inducible protein-protein interaction	Good internalization of the exosomes in leukemia cells and successful delivery of the miRNA21 loaded AS1411-Exos with significant induction of cellular apoptosis	[44]

## Data Availability

Not applicable.

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
