# Peer review of "Chemically Modified Extracellular Vesicles and Applications in Radiolabeling and Drug Delivery"

_pharmaceutics, 2022, doi:10.3390/pharmaceutics14030653_

Round 1
Reviewer 1 Report
I enjoyed reading the manuscript by N’Diaye et al. in order to improve the quality of the work for publication I would like to ask the author for the following major revisions:
- Clearly, the manuscript lacks any figures, many experiments, concepts and methodological explanation should be summarized as high quality figures. In particular the authors have emphasized on Click chemistry in different section of the manuscript and there is no illustration for this.
- The authors could still improve the reference section by citing the related articles/books in the literature.
- Many sentences are simply a summary of research in other articles. Authors should try to rewrite such sentences. For instance:
“In this context, Faruqu proposed a novel, reliable and 205 universal method for the radiolabeling of exosomes [25].”; “, it was concluded that this method is also efficient on EVs 316 with peptides and nanobodies targeting other receptors”; “The information provided in Maguire et al.'s study about the fusion of AAV with exosomes could further develop their use in medicine”; and any other sentences like these.”; “ Lee et al worked on some copper-free and strain-promoted azide[1]111 alkyne click (SPAAC) chemistry [17]. In this SPPAC reaction, the triazole linkage is formed”,
- I could find Few typos and errors in the manuscript. Latin words such as et al, ex vivo, in vitro, etc, e.g. … should be written in Italic format. “pH= #,#” should be changed to “pH #.#”; western blot should be “Western blot” (capital W). Abbreviations sometimes left undefined.
- Authors sometimes used informal language. Please go through the text and find a replacement: I could find the following sentences inappropriate for a scientific article: “damaged the extracellular vesicle”; “which gives big 262 hope in the progress of…”; “under 137 appropriate conditions.”; “Last but certainly not least is the activation of an acute hypersensitivity 39 reaction”; “in Maguire et al.'s study”; “it’s primordial to….”
- Some sentences are unclear, which means authors should carefully edit the grammar throughout the manuscript: “reduced 99mT-MDEs were 10-fold 173 more present in the liver than in the brain whereas for the intraperitoneal route, 2-fold.”; “to avoid taking too much capital on the 158 method and collecting better results, they set up a radiochemical radiolabeling of milk-derived 159 exosomes (MDE) with reduced 99mTc (IV)” ; “EV sub-populations”;” y, it was concluded that this method is”;” y, it was concluded that this method is”;
- The author should disclose whether the current manuscript is derived or will be going to be a thesis chapter.
- Table 5: “Hydrophobic insertion of cholesterol to improve the” should be more clarified.
Author Response
I enjoyed reading the manuscript by N’Diaye et al. in order to improve the quality of the work for publication I would like to ask the author for the following major revisions:
- Clearly, the manuscript lacks any figures, many experiments, concepts and methodological explanation should be summarized as high-quality figures. In particular the authors have emphasized on Click chemistry in different section of the manuscript and there is no illustration for this.
Response: Thank you for pointing this out. We agree with this comment. Therefore, we have added figures that summarize the principal methodologies mentioned in the revised manuscript.
- The authors could still improve the reference section by citing the related articles/books in the literature.
Response: We have accordingly improved the references section by including manuscripts recently published.
- Many sentences are simply a summary of research in other articles. Authors should try to rewrite such sentences. For instance: In this context, Faruqu proposed a novel, reliable and 205 universal method for the radiolabeling of exosomes [25].”; “, it was concluded that this method is also efficient on EVs 316 with peptides and nanobodies targeting other receptors”; “The information provided in Maguire et al.'s study about the fusion of AAV with exosomes could further develop their use in medicine”; and any other sentences like these.”; “ Lee et al worked on some copper-free and strain-promoted azide[1]111 alkyne click (SPAAC) chemistry [17]. In this SPPAC reaction, the triazole linkage is formed”.
Response: We agree with the reviewer's assessment; therefore, we revised the entire body text for English language and grammar; so we hope it now matches the journal standard.
- I could find Few typos and errors in the manuscript. Latin words such as et al, ex vivo, in vitro, etc, e.g. … should be written in Italic format. “pH= #,#” should be changed to “pH #.#”; western blot should be “Western blot” (capital W). Abbreviations sometimes left undefined.
Response: The requested modifications were modified.
- Authors sometimes used informal language. Please go through the text and find a replacement: I could find the following sentences inappropriate for a scientific article: “damaged the extracellular vesicle”; “which gives big 262 hope in the progress of…”; “under 137 appropriate conditions.”; “Last but certainly not least is the activation of an acute hypersensitivity 39 reaction”; “in Maguire et al.'s study”; “it’s primordial to….”
Response: Thank you for your comment regarding this aspect. As mentioned in point #3, we have performed a revision of the main body of the manuscript.
- Some sentences are unclear, which means authors should carefully edit the grammar throughout the manuscript: “reduced 99mT-MDEs were 10-fold 173 more present in the liver than in the brain whereas for the intraperitoneal route, 2-fold.”; “to avoid taking too much capital on the 158 method and collecting better results, they set up a radiochemical radiolabeling of milk-derived 159 exosomes (MDE) with reduced 99mTc (IV)” ; “EV sub-populations”;” y, it was concluded that this method is”;” y, it was concluded that this method is”;
Response: We thank the reviewer for these observations. As a result, the manuscript has been carefully revised, and most of the mentioned sentences have been rephrased to make the reading more comprehensive.
- The author should disclose whether the current manuscript is derived or will be going to be a thesis chapter.
Response: The current manuscript is not a thesis chapter.
Table 5: “Hydrophobic insertion of cholesterol to improve the” should be more clarified.
Response: We thank the reviewer for this suggestion. In order to improve the table 5 as requested, we have have added significant information in the paragraph so as in the table.
Reviewer 2 Report
It is a well written and comprehensive review about chemically modified extracellular vesicles and applications in radiolabeling and drug delivery. I recommend for publication after the following points are addressed.
- One figure about the overall picture of EVs for the applications in radiolabeling and drug delivery and one figure about the recent interesting work related to this topic are encouraged to be added.
- Line 131, '99mTc-HMPAO' should be '99mTc-HMPAO'. Please check all format issues.
- Several recent reviewers (doi.org/10.3390/pharmaceutics12050442; doi.org/10.1016/j.actbio.2020.06.036) related to EVs for drug delivery should be included. Meanwhile, more general ref. should be added as 44 ref. are not enough for a comprehensive review.
Author Response
It is a well written and comprehensive review about chemically modified extracellular vesicles and applications in radiolabeling and drug delivery. I recommend for publication after the following points are addressed.
- One figure about the overall picture of EVs for the applications in radiolabeling and drug delivery and one figure about the recent interesting work related to this topic are encouraged to be added.
Response: We agree with the reviewer’s remark, and as requested from reviwer#1, we have incorporated figures according to the review’s topic into the revised manuscript.
- Line 131, '99mTc-HMPAO' should be '99mTc-HMPAO'. Please check all format issues.
Response: Thank you for pointing this out. We have modified '99mTc-HMPAO' with '99mTc-HMPAO'into the revised version of the manuscript.
- Several recent reviewers (doi.org/10.3390/pharmaceutics12050442; doi.org/10.1016/j.actbio.2020.06.036) related to EVs for drug delivery should be included. Meanwhile, more general ref. should be added as 44 ref. are not enough for a comprehensive review.
Response: We have included recent reviews related to EVs in the revised manuscript as requested from the reviewer.

Reviewer 3 Report
The review manuscript entitled "Chemically modified extracellular vesicle and applications in radiolabeling and drug delivery" from N´Diaye et al involved the interesting bibliographical analysis regarding to the chemical approaches of conjugated EVs with radiotracers and drugs.
The introduction is corresponding to the topic of the manuscript, and it has bibliographical references to support the research.
Also, the authors analyzed the current advantages and disadvantages of these systems. In addition, the information they described is supported with clear tables that summarize the data.
Then, the chemical modifications for in vivo EVs were discussed considering:
- Covalent binding-method: in this item it would be important to analyze the potential cytotoxicity of the azide moiety when the systems is further metabolized (the authors reported accumulation in the liver and intestines).
- Several chemical reactions were reported such as SPAAC methods, radiolabeled (radiochemical), and BFC procedures. It is important to add the schematics procedures in order to help the reader to understand these methodologies.
- Please check the in vivo, in vitro, and ex vivo phrases (they should be in italics)
Also, chemical modifications on extracellular vesicle-mediates delivery cargo were reported:
- Regarding the suggestion for the above topic, it should be interesting to use some schematic images to explain the procedures the authors discussed in this item.
Besides, it is important to analyze the differences among this review manuscript and other similar publications in order to understand the potential utilization of the reported information in the present manuscript. Some relevant references (2019-2021) are cited below:
- Challenges for the development of extracellular vesicle-based nucleic acid medicines. Cancers 2021, 13, 6137. https://doi.org/10.3390/cancers13236137
- Drug delivery with extracellular vesicles: from imagination to innovation. DOI: 1021/acs.accounts.9b00109
- Extracellular vesicle-based drug delivery systems for cancer treatment. doi: 7150/thno.37097
- Modification of extracellular vesicles by fusion with liposomes for the design of personalized biogenic drug delivery systems. https://doi.org/10.1021/acsnano.8b02053
- Exploring interactions between extracellular vesicles and cells for innovative drug delivery system design. https://doi.org/10.1016/j.addr.2021.03.017
- Engineering exosome polymer hybrids by atom transfer radical polymerization. https://doi.org/10.1073/pnas.2020241118
- Liposomes and extracellular vesicles as drug delivery systems: a comparison of composition, pharmacokinetics, and functionalization. https://doi.org/10.1002/adhm.202100639
- A systematic review on the modifications of extracellular vesicles: a revolutionized tool of nano-biotechnology. https://doi.org/10.1186/s12951-021-01219-2
- Extracellular vesicles-based drug delivery system for cancer treatment. https://doi.org/10.1080/2331205X.2019.1635806
- Exosomes, a new star for targeted delivery. https://doi.org/10.3389/fcell.2021.751079
Moreover, the discussion covered all the previous analyses and some potential challenges, and it could be improved after the analysis of the suggested bibliographical references.
At this point, I encourage the authors to analyze the economical differences (which is the cheapest or which is the simplest methodologies to use according to each application) between all the reported methodologies would improve the discussion section. Additionally, this analyzes will allow the authors to enhance the conclusion section and future challenges.
I would like to invite the authors to add the abbreviation list of words at the end of this manuscript.
I recommend the acceptance of this manuscript after the authors performed the suggested corrections/additions.
Author Response
The review manuscript entitled "Chemically modified extracellular vesicle and applications in radiolabeling and drug delivery" from N´Diaye et al involved the interesting bibliographical analysis regarding to the chemical approaches of conjugated EVs with radiotracers and drugs.
The introduction is corresponding to the topic of the manuscript, and it has bibliographical references to support the research. Also, the authors analyzed the current advantages and disadvantages of these systems. In addition, the information they described is supported with clear tables that summarize the data. Then, the chemical modifications for in vivo EVs were discussed considering:
- Covalent binding-method: in this item it would be important to analyze the potential cytotoxicity of the azide moiety when the systems is further metabolized (the authors reported accumulation in the liver and intestines).
Response: The author personally thanks the reviewer for this advice. We therefore added a part discussing the cytotoxicity of copper in click chemistry.
- Several chemical reactions were reported such as SPAAC methods, radiolabeled (radiochemical), and BFC procedures. It is important to add the schematics procedures in order to help the reader to understand these methodologies.
Response: Thank you for this important suggestion. We have provided the schematics procedures relative to SPAAC methods, radiolabeled (radiochemical), and BFC procedures into the revised manuscript.
- Please check the in vivo, in vitro, and ex vivo phrases (they should be in italics)
Response: Thank you for pointing this out. As also requested from reviewer #1, we have modified them in Italics style into the revised version.
- Also, chemical modifications on extracellular vesicle-mediates delivery cargo were reported. Regarding the suggestion for the above topic, it should be interesting to use some schematic images to explain the procedures the authors discussed in this item.
Response: We highly appreciate the reviewer' insightful and helpful comments on our manuscript. As requested from the reviwer #1, we have added into the revised manuscript some schematic images making the manuscript more easily understandable for readers.
- Besides, it is important to analyze the differences among this review manuscript and other similar publications in order to understand the potential utilization of the reported information in the present manuscript. Some relevant references (2019-2021) are cited below:
1-Challenges for the development of extracellular vesicle-based nucleic acid medicines. Cancers 2021, 13, 6137. https://doi.org/10.3390/cancers13236137
2-Drug delivery with extracellular vesicles: from imagination to innovation. DOI: 1021/acs.accounts.9b00109
3-Extracellular vesicle-based drug delivery systems for cancer treatment. doi: 7150/thno.37097
4-Modification of extracellular vesicles by fusion with liposomes for the design of personalized biogenic drug delivery systems. https://doi.org/10.1021/acsnano.8b02053
5-Exploring interactions between extracellular vesicles and cells for innovative drug delivery system design. https://doi.org/10.1016/j.addr.2021.03.017
6-Engineering exosome polymer hybrids by atom transfer radical polymerization. https://doi.org/10.1073/pnas.2020241118
7-Liposomes and extracellular vesicles as drug delivery systems: a comparison of composition, pharmacokinetics, and functionalization. https://doi.org/10.1002/adhm.202100639
8-A systematic review on the modifications of extracellular vesicles: a revolutionized tool of nano-biotechnology. https://doi.org/10.1186/s12951-021-01219-2
9-Extracellular vesicles-based drug delivery system for cancer treatment. https://doi.org/10.1080/2331205X.2019.1635806
10-Exosomes, a new star for targeted delivery. https://doi.org/10.3389/fcell.2021.751079
- Moreover, the discussion covered all the previous analyses and some potential challenges, and it could be improved after the analysis of the suggested bibliographical references. At this point, I encourage the authors to analyze the economical differences (which is the cheapest or which is the simplest methodologies to use according to each application) between all the reported methodologies would improve the discussion section. Additionally, this analyzes will allow the authors to enhance the conclusion section and future challenges.
Response: We appreciate these important suggestions. However, even though, it would have been interesting to explore this aspect, However, it seems slightly out of scope because the aim of our review is to stress the importance of chemical modification approach enhancing the EVs functions.
I would like to invite the authors to add the abbreviation list of words at the end of this manuscript.
Response: Thank you for this suggestion. As requested from the reviewer, the number of abbreviations has been greatly decreased for easy reading. Some abbreviations have been replaced by more commonly used ones.
I recommend the acceptance of this manuscript after the authors performed the suggested corrections/additions.

Round 2
Reviewer 1 Report
thanks for addressing mycomments. you should be still adding more figures if you wish though. but it's acceptale from my side